# Identifying potential biases in code sequences in primary care electronic healthcare records: a retrospective cohort study of the determinants of code frequency

Thomas Beaney [1,2] Jonathan Clarke [2] David Salman [1,3]
Thomas Woodcock,[1] Azeem Majeed [1] Mauricio Barahona [2] Paul Aylin[1]

[1]Department of Primary Care and Public Health, Imperial College London, London, UK
[2]Department of Mathematics, Imperial College London, London, UK
[3]MSk Lab, Imperial College London, London, UK

**Correspondence to**
Dr Thomas Beaney;
thomas.beaney@imperial.ac.uk

## ABSTRACT

**Objectives** To determine whether the frequency of diagnostic codes for long-term conditions (LTCs) in primary care electronic healthcare records (EHRs) is associated with (1) disease coding incentives, (2) General Practice (GP), (3) patient sociodemographic characteristics and (4) calendar year of diagnosis.

**Design** Retrospective cohort study.

**Setting** GPs in England from 2015 to 2022 contributing to the Clinical Practice Research Datalink Aurum dataset.

**Participants** All patients registered to a GP with at least one incident LTC diagnosed between 1 January 2015 and 31 December 2019.

**Primary and secondary outcome measures** The number of diagnostic codes for an LTC in (1) the first and (2) the second year following diagnosis, stratified by inclusion in the Quality and Outcomes Framework (QOF) financial incentive programme.

**Results** 3 113 724 patients were included, with 7 723 365 incident LTCs. Conditions included in QOF had higher rates of annual coding than conditions not included in QOF (1.03 vs 0.32 per year, p<0.0001). There was significant variation in code frequency by GP which was not explained by patient sociodemographics. We found significant associations with patient sociodemographics, with a trend towards higher coding rates in people living in areas of higher deprivation for both QOF and non-QOF conditions. Code frequency was lower for conditions with follow-up time in 2020, associated with the onset of the COVID-19 pandemic.

**Conclusions** The frequency of diagnostic codes for newly diagnosed LTCs is influenced by factors including patient sociodemographics, disease inclusion in QOF, GP practice and the impact of the COVID-19 pandemic. Natural language processing or other methods using temporally ordered code sequences should account for these factors to minimise potential bias.

## STRENGTHS AND LIMITATIONS OF THIS STUDY

⇒ This study used a large and representative sample of patients in England, including 3 million patients with one of 208 incident diseases developed over 5 years.

⇒ We focused on incident diseases during the study period to minimise bias from historic or inactive diseases.

⇒ We found significant differences in the frequency of codes according to patient sociodemographics, GP practice and disease inclusion in Quality and Outcomes Framework, but could not determine whether these differences reflect differences in healthcare utilisation versus coding quality.

## BACKGROUND

Methods developed in natural language processing (NLP) are increasingly being employed to analyse routinely collected healthcare data, such as data recorded in the electronic healthcare record (EHR).[1–6] These methods show promise across a range of tasks, including prediction of health outcomes,[1 5 6] and clustering of co-occurring diseases.[2] Although developed for the analysis of language data, such as the free text data found in 'unstructured' medical records, NLP methods can also be applied to coded or 'structured' data found in many EHR databases. Using structured data, disease codes arranged in a temporal sequence in a patient's EHR history can be considered analogous to words in a sentence or document.[5]

In primary care EHRs, diagnostic codes may be entered either during a consultation, or entered outside, such as on receiving communication of a new diagnosis from hospital, or retrospectively coding a pre-existing diagnosis. In predictive modelling scenarios, such as those used in NLP, codes from both sources are relevant to understanding a patient's health status. However, a potential problem facing sequence-based methods is the extent to which repeated codes are an

**BMJ**

objective marker of a patient's health status and a presentation with a particular condition or relate to the quality of coding in the EHR.[7] Although previous studies of EHR data in England have shown the prevalence of many long-term conditions (LTCs) to be comparable to those from national statistics, these are often calculated based on the presence of a single diagnostic code.[8] Whether repeated codes for LTCs are entered in the EHR subsequently may be determined by a range of factors, including patient characteristics, clinician incentives and organisational policies, which may vary over time.[9 10]

Unlike in secondary care, where diagnostic coding directly impacts on payments, general practice in England receives funding primarily through capitated payments based on the size of the registered population,[11] with no direct financial incentive for code entry during a consultation. However, around 10% of funding comes from the Quality and Outcomes Framework (QOF), introduced in the National Health Service for GPs in 2004.[11] QOF provides financial incentives for meeting targets for a set of chronic conditions, including regular clinical reviews, and has been credited with improvements to data collection for these conditions.[12–14] Codes for conditions in QOF may occur more frequently than for conditions not included in the incentive scheme, which could affect sequence-based methods using recurrent codes.

Analytical methods using temporally ordered code sequences in the EHR may therefore be susceptible to biases in the frequency of codes entered following diagnosis, potentially resulting in models representing some people better than others. Awareness of the factors influencing the frequency of codes may help researchers using NLP methods by informing adjustment or sensitivity analyses. This study aims first to compare the frequency of repeated codes after diagnosis for a common set of LTCs. Second, we aim to determine whether the frequency of codes varies according to (1) disease inclusion in QOF, (2) GP practice, (3) patient sociodemographic characteristics and (4) calendar year of diagnosis.

## METHODS

### Data source

This study used data from the Clinical Practice Research Datalink (CPRD) Aurum dataset, which contains primary care data for GP practices using EMIS Web software.[15] We included all patients assessed by CPRD to be research acceptable (meeting certain quality criteria such as a valid registration date and date of birth[16]) with a continuous period of registration at a GP practice in CPRD between 1 January 2014 and 31 December 2020 (ie, without having deregistered in this period).[17] Patients were eligible if aged 18 years or over with at least one incident disease diagnosed between 1 January 2015 and 31 December 2019, allowing for at least one full year of practice registration before disease diagnosis and at least one full year of follow-up for each condition. Demographic data included age, sex, ethnicity and Index of Multiple Deprivation

(IMD) of the area in which the patient resided, grouped into deciles where 1 is the least deprived and 10 the most deprived.[18] Ethnicity is recorded as one of five categories, with recording in CPRD found previously to have high concordance with national estimates.[19] We focused on incident diseases to reduce the potential for confounding from historic conditions, some of which may no longer be active. Patients were followed up until the earliest of death, deregistration or the date of latest data extraction from their GP practice. Further information on the cohort structure is given in the online supplemental file (p2).

### Disease definitions

Diagnostic codes were extracted from the CPRD 'Observation' table and codes recorded during or outside of consultations were included. The date that the event occurred ('obsdate') was used, in preference to the date the code was entered. We included a total of 208 LTCs. These were defined based on a set of disease codes from Head *et al*, who selected 211 chronic conditions from 308 acute and chronic disease phenotypes developed from an earlier study.[20 21] We reviewed codes and made changes to the code lists for diabetes and added a new condition of 'chronic primary pain' (see online supplemental file p2-3). In the original code lists, conditions related to raised cholesterol or triglycerides are based only on laboratory results, rather than diagnostic disease codes. We excluded these conditions given that laboratory measurements may have different characteristics of coding frequency. Likewise, for obesity and chronic kidney disease, we used the diagnostic codes included in the code lists, but did not include body mass index and estimated glomerular filtration rate measurements. We considered a single code as diagnostic for each condition and defined the diagnosis date for each condition as the date of the earliest code for that condition. Diseases were stratified according to whether they appeared in QOF by two primary care clinicians, TB and DS (see online supplemental file p2-3).

### Statistical analysis

#### Descriptive statistics

For each disease newly diagnosed during the study period, we calculated the yearly number of subsequent codes (excluding the first code representing diagnosis) during follow-up:

$$y_i = \frac{\sum_{j=1}^{N} c_{i,j}}{\sum_{j=1}^{N} f_{i,j}}$$

where $y_i$ is the yearly number of codes following diagnosis for condition $i$, $c_{i,j}$ is the count of codes for condition $i$ in patient $j$ and $f_{i,j}$ is the number of years of follow-up for condition $i$ in patient $j$. T-tests were used to compare the mean yearly number of codes for QOF versus non-QOF conditions.

To examine variation in disease coding frequency by GP practice, we calculated, for each practice $k$, the mean number of codes per year for newly diagnosed diseases, $p_k$:

$$p_k = \frac{\sum_{j=1}^{N} \sum_{i=1}^{M} c_{i,j,k}}{\sum_{j=1}^{N} \sum_{i=1}^{M} f_{i,j,k}}$$

where $c_{i,j,k}$ is the count of codes for condition $i$ in patient $j$ in practice $k$, and $f_{i,j,k}$ is the number of years of follow-up for condition $i$ in patient $j$ in practice $k$. We then calculated the Pearson's correlation coefficient between the mean number of codes per year in each practice for QOF versus non-QOF conditions. We also compared the mean number of yearly codes in each practice stratified by the 2019 IMD decile of the GP practice. For conditions with at least 2 years of follow-up after the date of diagnosis, we calculated the ratio of the number of codes in the first year of diagnosis to the number of codes in subsequent years.

### Regression analyses

Data were formatted as panel data with patients measured over multiple calendar years (online supplemental table A1). We used mixed effects negative binomial regression to analyse the association between code frequency of newly diagnosed conditions in (1) the first year following diagnosis and (2) the second year following diagnosis, with patient factors and calendar year of diagnosis. We separated the outcome variable (code frequency) into first and second year after diagnosis due to preliminary analyses indicating significant differences over time. We also stratified the regression analyses by QOF inclusion, given our hypothesis that it may be an effect modifier of the relationships. To account for cases where a patient may have more than one QOF or non-QOF condition diagnosed within the same year, we averaged the code frequency for all newly diagnosed QOF or non-QOF conditions in each calendar year.

Included as covariates in the model were patient sociodemographic factors including age, sex, ethnicity and IMD decile of residence. We also included the count of QOF and non-QOF conditions for each patient. Due to small numbers, we excluded patients with gender recorded in CPRD as 'indeterminate' or with missing IMD deciles. Age and the count of QOF and non-QOF conditions were time-updated at the start of each calendar year, and other covariates were held fixed. We incorporated random effects for patient and fixed effects for calendar year as we wished to explicitly model the effect of time. Use of a Poisson model was considered, but the conditional variance was found to be significantly higher than the conditional mean ($p<0.001$) indicating a negative binomial to have better fit.[22] Model fit was assessed by calculating randomised quantile residuals, which indicated no departure from normality on quantile-quantile plots.[23 24]

For each regression model, we calculated the predicted count of disease codes for each patient per year and then calculated the mean for each GP practice. This indicated that significant variation remained in the mean counts according to GP practice (online supplemental figure A1). We therefore incorporated fixed effects for GP practice within the regression models to account for practice-level variation (see online supplemental file p5 for model equation). We also compared the Akaike Information Criteria (AIC) of models with and without practice fixed effects.

To assess whether code frequency was a function of overall number of primary care consultations, we conducted a sensitivity analysis including average number of yearly consultations (irrespective of condition recorded in the consultation) in year 1 or year 2 added as a covariate into the main regression models (categorised into <1, 1–2, 3–4, 5–9 or 10 or more consultations in the year). Python V.3.10.6 and Pandas V.1.4.3 were used in data processing and plots and Stata V.17.0 and R studio V.4.2.1 were used for regression analyses.

### Patient and public involvement

This research programme is supported by a patient and public advisory group who fed back to the researchers on the diseases included in the study but were not directly involved in this study.

## RESULTS

A total of 6 174 115 patients aged 18 years or over and with a continuous registration period between 1 January 2014 and 31 December 2020 were eligible for inclusion in the study. Of these, 3 113 724 (50.4%) had at least one incident disease diagnosed between 1 January 2015 and 31 December 2019. Characteristics of the eligible population are shown in table 1; 21.4% of patients were aged between 18 and 40 years as of the study start date, and 7.0% were aged 80 years or over. There were more women than men (54.1% vs 45.9%), most (76.7%) were of White ethnicity and there were relatively more patients in less deprived IMD deciles (51.7% in the least deprived half). Of patients with pre-existing conditions developed before the study start date, 31.6% had one or more QOF conditions and 71.3% had one or more non-QOF conditions. Hypertension was the most prevalent pre-existing condition (24.1%), and the frequencies of all pre-existing conditions are shown in the online supplemental table A2. The 3 060 391 patients who were not eligible (as they did not develop an incident disease over the study period) were more likely to be younger and more likely to be male than those eligible (online supplemental table A3).

### Code frequency by disease and by time from diagnosis

A total of 7 723 365 diseases were diagnosed during the study period with follow-up times for each disease ranging from 1.0 to 7.2 years (mean 4.1 years). There was substantial variation in the yearly code frequency after diagnosis

**Table 1** Sociodemographic characteristics of patients included in the study

| Patient characteristic | Total | Percent (%) |
|---|---|---|
| Age (years) | | |
| 18–39 | 665 543 | 21.4 |
| 40–49 | 562 934 | 18.1 |
| 50–59 | 604 284 | 19.4 |
| 60–69 | 585 062 | 18.8 |
| 70–79 | 476 626 | 15.3 |
| 80+ | 219 275 | 7.0 |
| Gender | | |
| Female | 1 684 942 | 54.1 |
| Male | 1 428 734 | 45.9 |
| Indeterminate | 48 | <0.1 |
| Ethnicity | | |
| White | 2 388 332 | 76.7 |
| South Asian | 194 477 | 6.2 |
| Black | 103 504 | 3.3 |
| Other | 36 430 | 1.2 |
| Mixed | 27 572 | 0.9 |
| Missing | 363 409 | 11.7 |
| IMD decile | | |
| 1 (least deprived) | 358 948 | 11.5 |
| 2 | 320 042 | 10.3 |
| 3 | 320 340 | 10.3 |
| 4 | 323 782 | 10.4 |
| 5 | 287 114 | 9.2 |
| 6 | 303 798 | 9.8 |
| 7 | 304 044 | 9.8 |
| 8 | 298 185 | 9.6 |
| 9 | 305 563 | 9.8 |
| 10 (most deprived) | 290 214 | 9.3 |
| Missing | 1694 | 0.1 |
| Pre-existing QOF conditions* | | |
| 0 | 2 130 680 | 68.4 |
| 1 | 393 905 | 12.7 |
| 2 | 224 147 | 7.2 |
| 3 | 142 104 | 4.6 |
| 4 or more | 222 888 | 7.2 |
| Pre-existing non-QOF conditions* | | |
| 0 | 893 765 | 28.7 |
| 1 | 561 300 | 18.0 |
| 2 | 506 053 | 16.3 |
| 3 | 386 912 | 12.4 |
| 4 or more | 765 694 | 24.6 |
| Total | 3 113 724 | |

Continued

**Table 1** Continued

| Patient characteristic | Total | Percent (%) |
|---|---|---|

*Pre-existing conditions defined as of study start date.
IMD, Index of Multiple Deprivation; QOF, Quality and Outcomes Framework.

for each condition diagnosed during the study period. Diabetes (types 1, 2 and unspecified), polymyalgia rheumatica, motor neurone disease and dementia had the highest median number of codes per year (online supplemental table A4). For many chronic diseases, yearly code frequency was low, for example, only 5% of patients with spina bifida had ≥0.5 codes per year. Conditions included in QOF on average had significantly higher mean number of yearly codes (1.03) than conditions not included in QOF (0.32; p<0.0001).

The number of codes was higher in the first year after diagnosis than in subsequent years for almost all conditions, except for secondary bowel or pleural malignancy and diabetic eye disease, for which code frequency was higher on average after the first year of diagnosis. QOF conditions on average had lower ratios of codes in the first compared with subsequent years than non-QOF conditions (4.8 vs 5.7 times higher in year 1). However, diseases representing major cardiovascular events, such as myocardial infarction, were coded much more frequently in the first year from diagnosis than in subsequent years (online supplemental figures A2 and A3).

### Variation in coding frequency by GP practice
There was a wide range in the mean yearly number of codes per condition between GP practices, with higher code frequency for QOF compared with non-QOF conditions (online supplemental figure A4). There was a strong correlation (r=0.88) between GP practice mean code frequency for QOF and non-QOF conditions, indicating that those practices with high code frequency for QOF conditions also had high code frequency for non-QOF conditions (figure 1). There was no observed trend according to the GP practice-level IMD decile (online supplemental figure A5).

We calculated the expected counts of codes for new diseases in year 1 and year 2 following diagnosis, predicted from negative binomial regression models. Expected mean counts per condition at GP practice level showed substantially less variation compared with the observed mean counts for both QOF and non-QOF conditions in year 1 and year 2 (online supplemental figure A1) indicating substantial residual practice-level variation independent of patient sociodemographic factors.

### Variation in disease frequency by sociodemographics and over time
We found significant associations between code frequency in year 1 and year 2 following diagnosis with patient sociodemographic factors and calendar

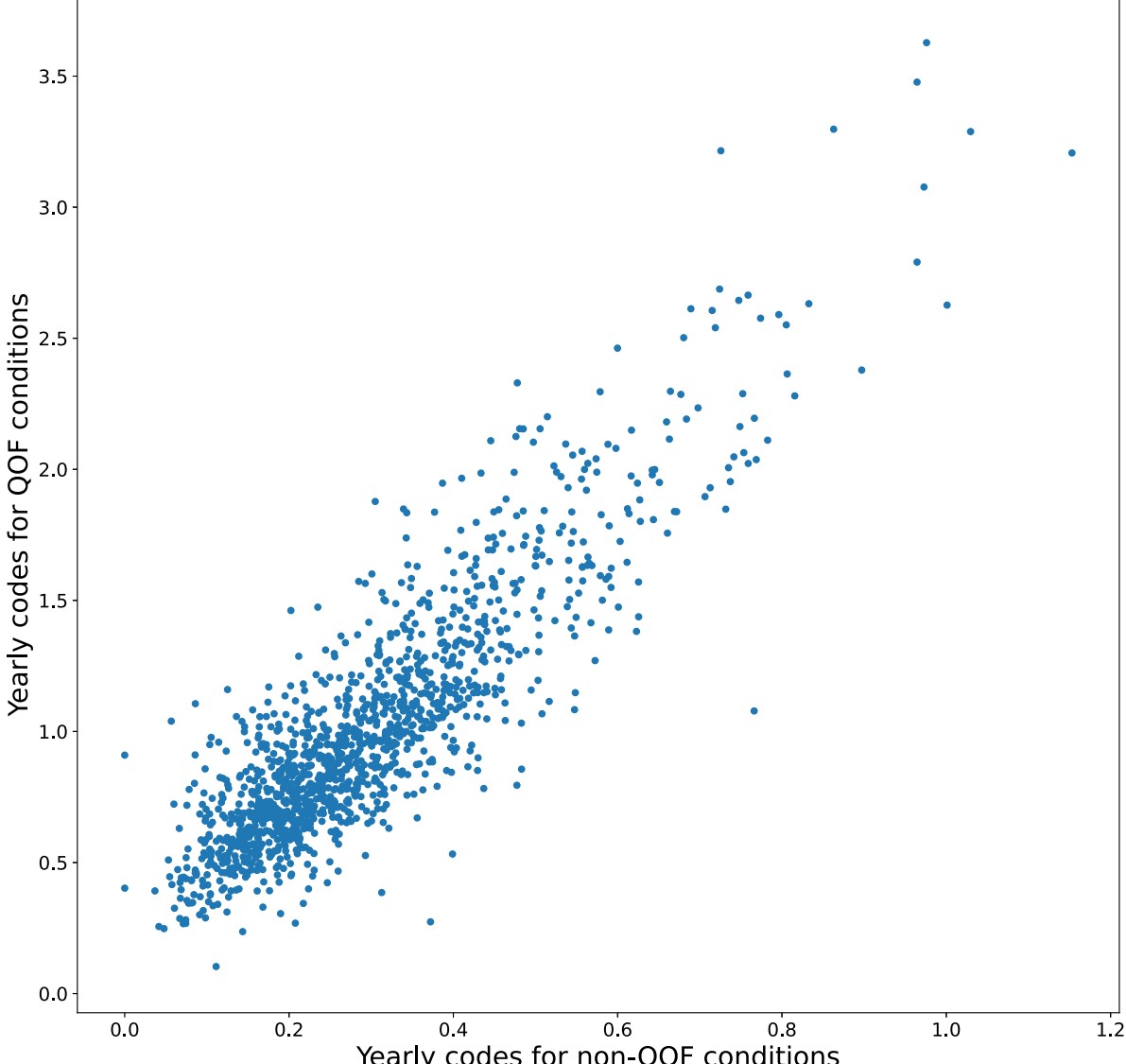

**Figure 1** Scatter plot of mean yearly number of codes following diagnosis for QOF versus non-QOF conditions for each GP practice. Different ranges used in each axis. QOF, Quality and Outcomes Framework.

year of diagnosis for both QOF and non-QOF diseases from mixed effects negative binomial regression, after adjustment for number of pre-existing conditions (figures 2 and 3, and online supplemental tables A5–A8). Inclusion of GP practice fixed effects in the regression models resulted in very similar coefficients for patient sociodemographic factors, and a significantly lower AIC indicating better model fit and so results are presented including practice-level effects.

### Associations with QOF conditions
Younger patients tended to have a higher frequency of codes in the first year following diagnosis compared with older patients (figure 1). However, in the second year from diagnosis, there was a U-shaped relationship with age, with the youngest and oldest age groups having the lowest rate of codes. Males had on average a small 3% increase (95% CI 1.03 to 1.03) in the

incidence rate of codes in year 1 and 11% (95% CI 1.11 to 1.12) increase in year 2 compared with females. There was a strong relationship with ethnicity, with people of non-white ethnicities having lower rates of code frequency than people of white ethnicity in year 1, but higher rates in year 2. There was a strong trend towards higher code frequency in year 1 and year 2 with increasing levels of deprivation.

### Associations with non-QOF conditions
For conditions not included in QOF, relationships were more consistent across year 1 and year 2 following diagnosis (figure 2). The 18–40 years age group had the highest rate of codes in both year 1 and year 2, with only small differences between other age groups. There was no difference in the rate of codes in males and females in year 1, but males had a lower rate of codes in year 2. Lower rates of codes were found in people of non-white

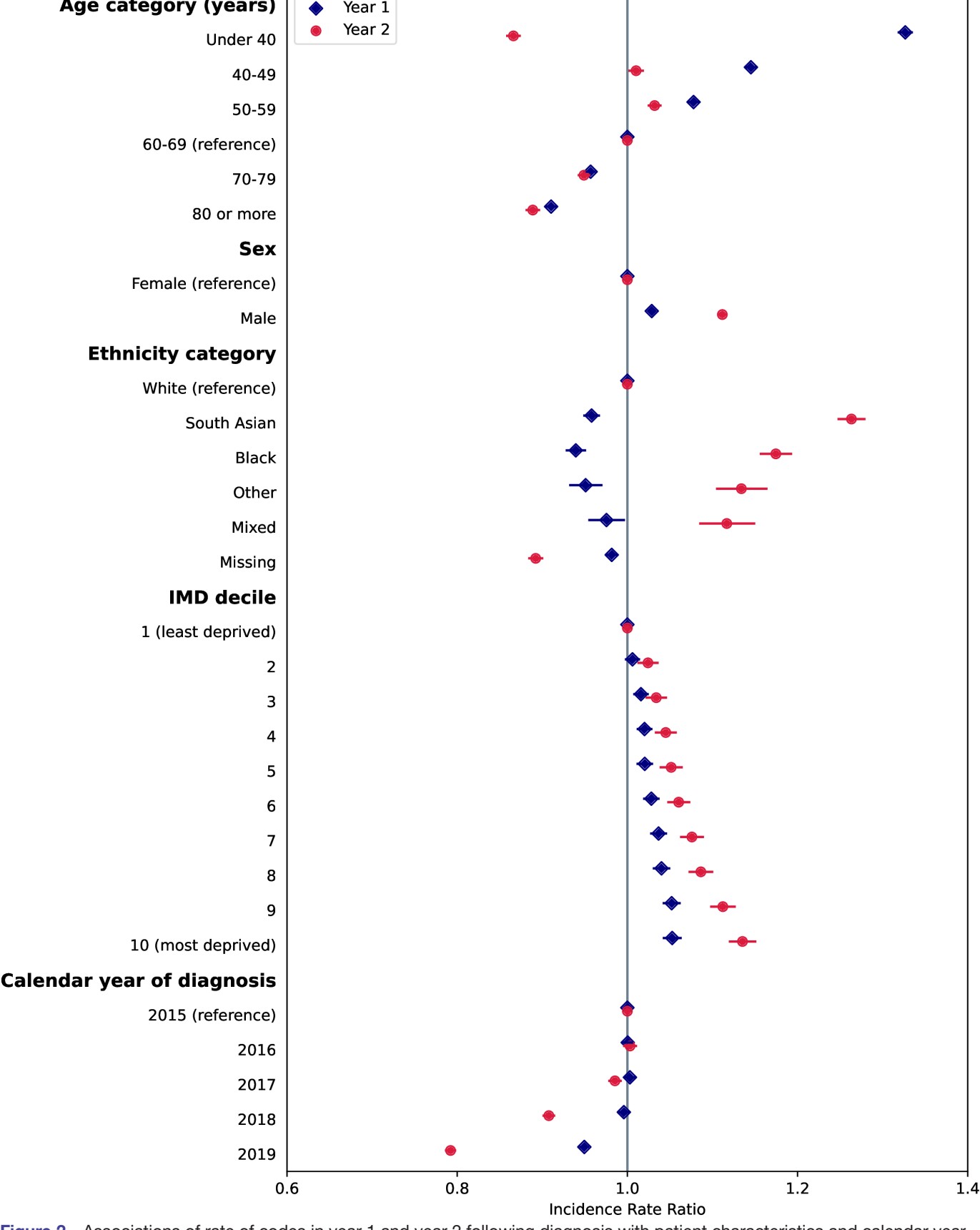

**Figure 2** Associations of rate of codes in year 1 and year 2 following diagnosis with patient characteristics and calendar year, for conditions included in the Quality and Outcomes Framework (QOF). Points represent estimates of the incidence rate ratio and bars represent 95% CIs from negative binomial regression models. Corresponding values and coefficients for pre-existing QOF and non-QOF conditions are given in online supplemental tables A5 and A6. IMD, Index of Multiple Deprivation.

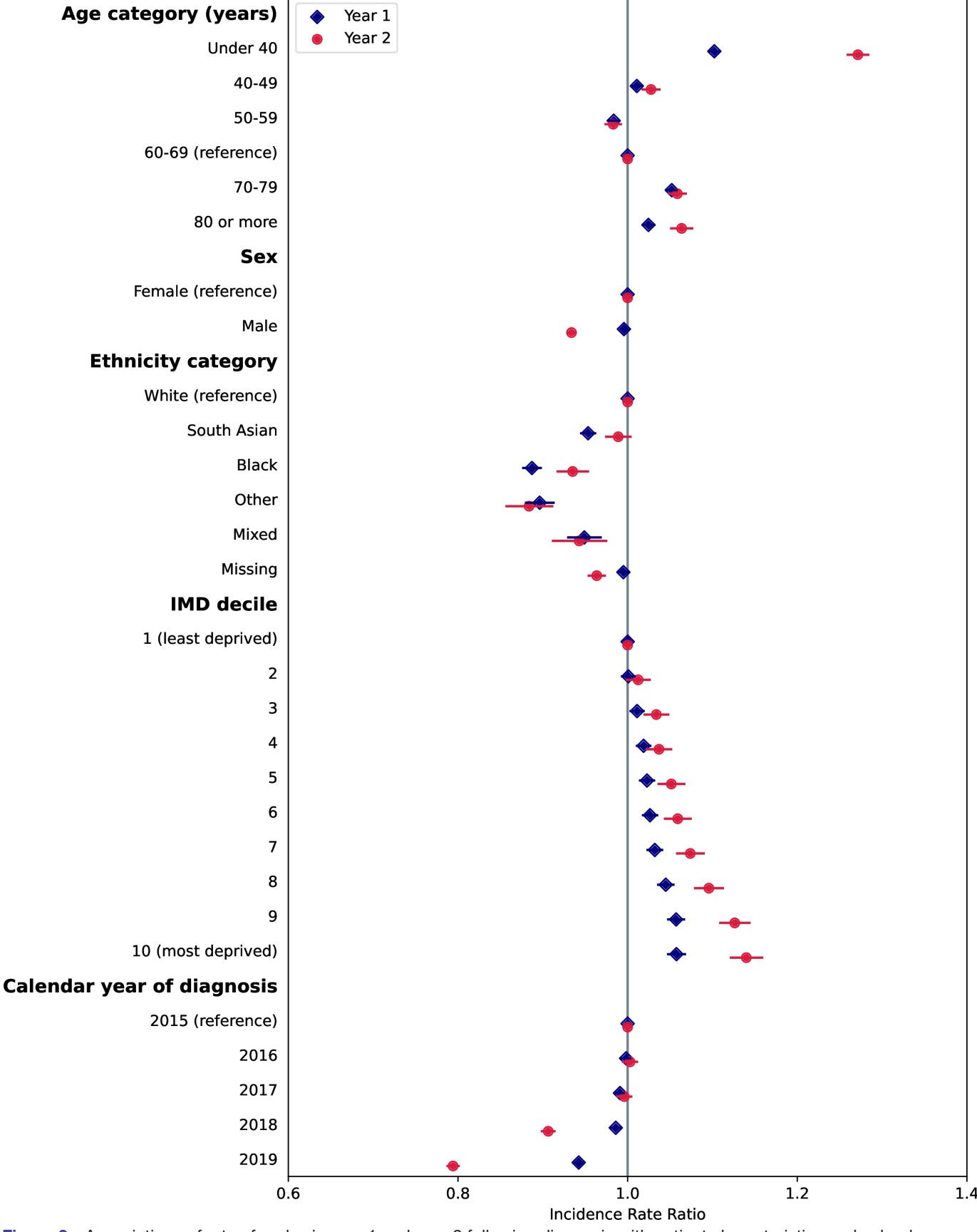

**Figure 3** Associations of rate of codes in year 1 and year 2 following diagnosis with patient characteristics and calendar year, for conditions not included in the Quality and Outcomes Framework (QOF). Points represent estimates of the incidence rate ratio and bars represent 95% CIs from negative binomial regression models. Corresponding values and coefficients for pre-existing QOF and non-QOF conditions are given in online supplemental tables A7 and A8. IMD, Index of Multiple Deprivation.

ethnicities compared with people of white ethnicity, except for South Asian ethnicity in year 2. Similar to QOF conditions, there was a strong trend towards higher code rates in year 1 and year 2 with increasing deprivation.

## Associations with calendar year

For both QOF and non-QOF conditions, code rates were similar for conditions diagnosed in 2016 and 2017 compared with 2015 (figures 1 and 2). For codes in year 1, rates for conditions diagnosed in 2018 were similar to 2015, but rates for diseases diagnosed in 2019 were 5% and 6% lower than 2015 for QOF and non-QOF conditions, respectively. For codes in year 2, rates were significantly lower in 2018 (9% and 9% lower for QOF and non-QOF, respectively) and 2019 (21% and 21% lower for QOF and non-QOF, respectively) compared with 2015.

## Adjustment for total number of consultations

A sensitivity analysis was used to adjust for total number of consultations in year 1 or year 2 from diagnosis (online supplemental tables A5–A8). Total number of consultations in each year were strongly linked to the rate of codes. For newly diagnosed QOF conditions, the associations with age, sex and ethnicity in years 1 and 2 remained significant after adjustment (online supplemental tables A5 and A6). However, the association with deprivation was attenuated, although there remained an association with higher rates of codes with higher deprivation in year 2. For newly diagnosed non-QOF conditions, after adjustment for consultations, age and ethnicity remained significantly associated, but males had significantly higher rates of codes than females (online supplemental file 1). Associations with deprivation were attenuated, but there remained a small but significant association in year 2.

## DISCUSSION

With an increased use of NLP methods incorporating temporally ordered code sequences in the primary care EHR, we need to better understand the frequency and determinants of repeated occurrences of diagnostic codes. Our study demonstrates significant associations in the frequency of codes for newly diagnosed conditions according to patient sociodemographic factors, GP practice, disease inclusion in QOF and calendar year. We are unable to fully assess the extent to which the relationships in our study are explained by the quality of coding, or by how patients use healthcare services for a particular condition. However, a sensitivity analysis adjusting for total number of yearly consultations per patient yielded similar results, suggesting that variation in coding quality is likely to play a role. Our findings have implications for researchers using code sequences, emphasising the importance of considering these factors as potential sources of bias.

## Patient sociodemographics

Patient characteristics including age, sex and ethnicity were strongly linked to code frequency, although associations were inconsistent across QOF and non-QOF conditions, and for QOF conditions, were not consistent across the first and second year following diagnosis. People of non-White ethnicity, for example, had lower code rates for QOF conditions in year 1, but higher in year 2, compared with people of White ethnicity. We found consistent patterns with deprivation, with higher code frequency in people living in more deprived areas, which might be explained by greater healthcare needs and more healthcare visits.[25] A sensitivity analysis adjusting for the total number of consultations attenuated the association with deprivation, suggesting that the relationship of code frequency with deprivation was partially, but not fully, explained by total primary care contacts. These findings likely point to differences in the mix of conditions between patient groups, greater healthcare need or healthcare-seeking behaviours, or greater likelihood of coding by GPs when presenting for appointments.

## GP practice

Substantial variation was found in the frequency of codes between GP practices, which persisted after accounting for differences in patient mix in terms of age, sex, deprivation, ethnicity, number of chronic conditions and in year of diagnosis. Although this may indicate unmeasured confounding in the characteristics of patients between practices, it likely represents policies and practices that influence coding which vary between organisations and clinicians.[9] For example, some GP practices may be more rigorous about coding data in clinical consultations and in correspondence from specialist services on diagnoses made in secondary care. Previous research has suggested that clinicians are more similar to those in the same practice than they are to clinicians in different practices with respect to treatment and diagnostic decisions.[26] Variation between clinicians in coding practices is likely to be significant both within and between practices, but this information was not accessible for the study, and its analysis would introduce multiple hierarchical dependencies outside the scope of this work. Future work could consider individual clinician effects on coding practices in the EHR.

## QOF and non-QOF conditions

Code frequency was significantly higher for conditions included in QOF compared with conditions not included. Previous research has highlighted changes to policies and procedures within GP practices to meet targets, including improved disease registries, which may lead to an increased likelihood of a code being entered for a given condition. We found substantial variation between GP practices in the mean code frequency for QOF conditions, but interestingly, this was strongly correlated (r=0.88 and figure 1) with code frequency for non-QOF conditions, suggesting that practice-level effects impact on coding across all conditions, rather than specifically those incentivised by

QOF. However, it is not possible in our study to determine whether differences in code frequency between QOF and non-QOF conditions are explained by greater healthcare need or an increased number of healthcare contacts for QOF conditions, or are explained by higher likelihood of a condition being coded when a patient presents.

## Calendar year

Accounting for calendar time in analyses of patient trajectories is a methodological concern, as the further back in time in the medical record, particularly before the advent of the EHR and QOF, the greater the chance that coding practices, and even disease categories, vary.[27] Although our study started relatively recently in 2015, and we cannot infer code frequency before this time, we found consistency in code frequency over a short time-span from 2015 to 2017. The decline in year 1 codes in 2019, and year 2 codes in 2018 and 2019 likely relates to the impact of the COVID-19 pandemic which impacted significantly on health services in England from March 2020.[28] Previous studies have shown reductions in patients presenting with particular conditions, and a reduction in appointment numbers in primary and secondary healthcare in England. Analyses reliant on coding frequency should therefore consider using calendar year in addition to patient age in modelling patient trajectories, or limiting analyses to defined time period.

## Strengths and limitations

A strength of our study is the inclusion of a large number of patients from a representative sample of primary care in England, which makes our findings generalisable to the national population.[15] We included only patients with newly incident diseases to minimise potential confounding from diseases diagnosed historically, some of which might no longer be active. We also only included patients with continuous follow-up over the study period and with at least 1 year of full practice registration to reduce bias from overestimation of incidence immediately following registration.[17] We also excluded patients who died less than <1 year from a new diagnosis, which may impact on disease frequency estimates for disease which have poor survival. We considered using annualised rates for those with less than a full year of follow-up, but this resulted in very high annualised counts for some individuals with short follow-up and might introduce additional bias if patients were to seek out care in advance of re-registering at another GP practice.

Our study has focused on structured healthcare data, whereas much of the consultation is recorded as unstructured 'free-text'.[7] Although unstructured primary care data contain much richer information on the details of a presentation that may not be fully reflected in the coded entries, this information is not currently available from CPRD, but research in future could examine the agreement between structured and unstructured primary care EHR data. This would allow a more robust estimation of the content and diseases covered during a consultation.

We stratified conditions according to QOF status given our hypothesis that it may influence coding frequency. However, we also found variation within categories, for example, polymyalgia rheumatica and motor neurone disease, which are not included in QOF, had high number of yearly codes, whereas cardiovascular events such as transient ischaemic attack, included in QOF, had low yearly codes. Given the general, comparative nature of this paper, and its aim to examine relationships over many conditions, a condition-specific analysis of coding frequency was out of scope.

## Implications for researchers

Our findings have implications for researchers using code sequences recorded in primary care structured data. The frequency of repeated diagnostic codes relates to patient-specific and condition-specific factors, coding incentives and practice-level factors. Although we cannot determine if these findings represent disease burden and healthcare need, it is likely that biases in coding operate at various levels. Specific approaches to reduce the impact of bias will depend on the methodology, but our work does suggest general principles.

First, to consider the potential for bias within the data source and whether stratification may reduce it, for example, by selecting a smaller number of healthcare organisations or a narrower time period. Second, to consider adjustment or inclusion of patient, condition, GP practice and calendar year variables within analytical models. However, such an approach is not always recommended, particularly if prediction is the aim, as inclusion of factors such as ethnicity in algorithms may reinforce existing bias.[29] In NLP, text style transfer is often used as a method to control for different styles of writing, which may have relevance to approaches to account for the different coding styles of clinicians.[30] However, these approaches are complicated within the EHR as people are likely to see multiple different clinicians over time, with a small set of codes recorded at each visit. Finally, it is vital that generated representations or predictions from modelling are evaluated in different patient subgroups.

## Implications for clinical practice

Although difficult to determine the extent to which our findings are attributed to coding quality versus healthcare utilisation, previous studies have reported variability in coding across practices for specific conditions.[31 32] This highlights a need to improve the quality of coding in primary care, given its impact on the reliability and usefulness of the data for secondary purposes including research. Improving the quality of coding in primary care poses several challenges, due to the different incentives for clinicians, who document most of the consultation in free text.[7] Potential strategies include implementing structured templates for recording consultations, or developing NLP methods capable of interpreting and codifying the free text documented during clinical encounters, without adding to clinician workload.[7]

## CONCLUSION

Our study found significant variation in the frequency of diagnostic codes recorded in the primary care EHR after diagnosis, related to patient sociodemographics, coding incentives and GP practice, and a significant reduction in the frequency of codes associated with the onset of the COVID-19 pandemic. These factors should be considered by researchers using NLP methods, or other approaches using temporally ordered sequences of codes in primary care EHRs, to reduce the risk of bias.

**Acknowledgements** Data management was provided by the Big Data and Analytical Unit (BDAU) at the Institute of Global Health Innovation (IGHI).

**Contributors** TB conceptualised the study, conducted the data management and formal analysis and wrote the first draft of the manuscript. TB, JC, DS, TW, AM, MB and PA contributed to the study design, methodology, interpretation of findings and reviewing and editing the manuscript. TB is the guarantor and accepts full responsibility for the work and the conduct of the study, had access to the data and controlled the decision to publish. The corresponding author attests that all listed authors meet authorship criteria and that no others meeting the criteria have been omitted.

**Funding** This research is funded through a clinical PhD fellowship awarded to TB from the Wellcome Trust 4i programme at Imperial College London. JC acknowledge support from the Wellcome Trust. MB acknowledges support from EPSRC grant EP/N014529/1 supporting the EPSRC Centre for Mathematics of Precision Healthcare. TW, AM and PA acknowledge support from the National Institute for Health and Care Research (NIHR) under the Applied Research Collaboration (ARC) Northwest London.

**Disclaimer** The views expressed in this publication are those of the authors and not necessarily those of the NHS, the NIHR, the Wellcome Trust or the Department of Health and Social Care.

**Competing interests** None declared.

**Patient and public involvement** Patients and/or the public were involved in the design, or conduct, or reporting, or dissemination plans of this research. Refer to the 'Methods' section for further details.

**Patient consent for publication** Not applicable.

**Ethics approval** Data access to the Clinical Practice Research Datalink (CPRD) and ethical approval was granted by CPRD's Research Data Governance Process on 28 April 2022 (protocol reference: 22_001818).

**Provenance and peer review** Not commissioned; externally peer reviewed.

**Data availability statement** Data may be obtained from a third party and are not publicly available. The data used in this study are not publicly available as access is subject to approval processes. Further information on data access is available from CPRD: https://cprd.com/research-applications.

**ORCID iDs**
Thomas Beaney http://orcid.org/0000-0001-9709-7264
Jonathan Clarke http://orcid.org/0000-0003-1495-7746
David Salman http://orcid.org/0000-0002-1481-8829
Azeem Majeed http://orcid.org/0000-0002-2357-9858
Mauricio Barahona http://orcid.org/0000-0002-1089-5675

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
