## [Reviewer comments · BMJ Open]

ARTICLE DETAILS

TITLE (PROVISIONAL)	Identifying potential biases in code sequences in primary care electronic healthcare records: a retrospective cohort study of the determinants of code frequency
AUTHORS	Beaney, Thomas; Clarke, Jonathan; Salman, David; Woodcock, Thomas; Majeed, Azeem; Barahona, Mauricio; Aylin, Paul

VERSION 1 – REVIEW

REVIEWER	Verheij, Robert A NIVEL, Netherlands Institute for Health Services Research
REVIEW RETURNED	02-Apr-2023

GENERAL COMMENTS	Main comments Although interesting at first sight, I have a number of problems with this paper: - At first sight this paper seems to address data quality issues. But after a while I wasn't so sure about that. It was not clear to me what this paper is about. Is it about health care utilization or is it about coding habits of gp practices?- If it is about health care utilization: There is an abundance of literature on health care utilization (numbers of consultations) in relation to socioeconomic differences, practice variation and qof. So if that is what this is about, the results are far from new.- If it is about coding habits, the authors should give us more information about the link between health care and HER record keeping. Is it possible in the EMIS EHR system to record consultations without diagnosis codes? It possible that codes have been recorded without contacts/consultations? can there be contacts/consultations without codes. The methods section suggests that this is possible, suggesting the paper is about coding habits rather than consultations. But if it is, why not simply calculate diagnosis code / consultation ratio's.- One other problem with this paper is that there is no clear description of the problem addressed and why it is important.- A clear research question is lacking. As are clear statements regarding the hypotheses to be tested.- The paper lacks a clear description of the types of study or study purposes for which the coding issue is relevant. Minor comments: • Variation in referral rates is likely to impact the occurrence of codes/consultations.• What are the implications of the study for Emis?• In several places it is suggested that patients are actually diagnosed in general practice. Very often this is not true. For many diseases patients are exclusively diagnosed in hospital.• Disease coding incentives: are they not incentives for providing care rather than coding.
--

	 • Selection of people who did not move house for two years, and who remained in the same practices. This will result in bias in the data studied. Has this effects on the results of the study? • What about completeness of ethnicity data. Is this recorded in EHRs? How complete and reliable is that information? • Not made clear how qof is related to coding rather than to consultations. • I don't understand how to interpret the scatterplots. If one disease has more coded diagnoses in a subsequent year after first incident consultation than another code. What does that mean? All in all, this may in the end turn out to be a very interesting paper with very relevant analyses, and I would advise the authors to solve the problems described and resubmit. Electronic health records data are currently being used for very many purposes. Epidemiological studies, health care utilization studies, health care quality studies, access to care studies etcetera. Possible biases might render the data unfit for some or all of these purposes and papers addressing such biases are important. All too often researchers tend to think that the data speak for themselves. I would like to encourage the authors of this paper to review their work carefully and then resubmit.
--	---

REVIEWER	Ranard, Benjamin Columbia University Irving Medical Center, Division of Pulmonary, Allergy, and Critical Care Medicine, Department of Medicine
REVIEW RETURNED	20-Apr-2023

GENERAL COMMENTS	In this interesting study, the authors analyzed the diagnostic codes from a large group of outpatients in England. The stated rationale for the study was to identify factors affecting diagnostic coding of new diseases in order to inform future studies of sequences of diagnostic codes given to individual patients. The authors showed that the frequency of diagnostic coding for new diseases was associated with patient sociodemographics, incentivization by the Quality and Outcomes Framework (a financial incentive program), GP practice, and the onset of COVID-19. This study has the potential to be useful in the future both for analyzing sequences of diagnostic coding and for health services research in general that aims to take advantage of diagnostic-code based data. Major Comments  1. Incident diseases are diagnosed by allowing at least one full year of practice registration before incident disease diagnosis. However, younger people (say 25-year-olds) seem less likely to visit their GP at least annually. Patients who visit their GP less frequently are more likely to be diagnosed with an incident disease when they do eventually visit their GP (since all of their prior comorbid diseases haven't been recorded in the study period). Am I correct in thinking that diseases associated with patients who visit their GP infrequently are more likely to be diagnosed as an "incident disease" when they do finally visit their GP? 2. Do we already know from other literature, or is there any analysis you can do, that would tell us how representative these diagnostic codes are of clinical reality? For example, is your incidental rate of diabetes using diagnostic codes at all similar to a more gold standard known rate of diabetes incidence in England? It would be helpful to know how well these incident rates of diagnostic codes are associated with the clinical reality.
--

	Minor Comments  1. In the United States, diagnostic codes are mainly used as billing codes. Additional codes are added to encounters to justify the complexity of the medical visit for billing purposes. For providers in an Accountable Care Organization or some sort of value-based payment model, documenting additional codes may be incentivized to ensure providers receive "credit" for the patient's level of sickness or comorbidity. Additionally, ICD codes may be necessary for prescribing medications. Adding a line or two about how/why providers code in England (besides for QOF incentives) would be useful background for international readers. 2. Page 6 line 19: what is a research acceptable patient? Is this a patient who has indicated their medical record can be used in research? 3. Page 6 line 19: what is required to maintain "a continuous period of registration at a GP practice?" Does this mean the patients visited their GP at least in a certain frequency or is it more like the patient was signed up for a GP as of 1/2014 and they never removed themselves from the patient panel? 4. Appendix page 2. Can you add a justification for: "We reviewed the codes in these lists, and made amendments to the code lists for diabetes, to remove Type 1 and Type 2 codes from the other/unspecified code list." 5. Page 6, line 55-56: Did the two primary care clinicians have perfect agreement or what was the Kappa? Not essential, but it would be nice to know the agreement or whether consensus was used. 6. Page 10: can you add in the appendix a "table 1" that compares the sociodemographic characteristics of patients included in the study with the 50% of patients who were excluded due to not having one incident disease in the study period? Is it possible to add a few clinical characteristics to both groups (x% had a diabetes code, y% had a heart failure code, etc)? This would not be incidence but whether the patient has ever had a code in order to get a sense of the patient population. 7. Page 11: The Figure 1 legend is erroneously showing up on my PDF on page 11. And figure legend 2 and 3 are showing up on page 13 of PDF. 8. Page 15 line 14 looks like it may be a typo.
--	---

VERSION 1 – AUTHOR RESPONSE

Reviewer comments:

Reviewer: 1

Dr. Robert A Verheij, NIVEL, Netherlands Institute for Health Services Research

Comments to the Author:

Main comments

Although interesting at first sight, I have a number of problems with this paper:

- At first sight this paper seems to address data quality issues. But after a while I wasn't so sure about that. It was not clear to me what this paper is about. Is it about health care utilization or is it about coding habits of gp practices?

Author reply: Thank you for highlighting this issue, which is difficult to disentangle. For a given disease/condition, we are unable to assess (without free text data, which is not available) whether the absence of a code is related to a patient not presenting with that

condition, or that they presented with a condition, but it was not coded. Furthermore, codes can be entered independent of consultations, for example from hospital letters, and these are important to include (as we did here) when determining a patient's health status. As a result, our findings may be explained both by differences in how patients present (i.e. utilisation) and by differences in how diseases are coded (i.e. data quality).

We have made several changes to the manuscript to acknowledge and address your concerns. The introduction has been substantially rewritten to more clearly highlight that we are including codes that occurred within or outside of a consultation and to specifically address the types of studies that we are concerned with (those using temporally-ordered sequences of codes). We also come back to the challenge of distinguishing healthcare utilisation from data coding quality in the first paragraph of discussion. Although we cannot fully address the problem, we did conduct a sensitivity analysis, presented at the end of Results, accounting for total primary care contacts over the year, which yielded similar results. This suggests that there are effects independent to patient presentation/utilisation. We have added the following to the first paragraph of Discussion:

“We are unable to fully assess the extent to which the relationships in our study are explained by the quality of coding, or by how patients use healthcare services for a particular condition. However, a sensitivity analysis adjusting for total number of yearly consultations per patient yielded similar results, suggesting that variation in coding quality is likely to play a role. Our findings have implications for researchers using code sequences, emphasising the importance of considering these factors as potential sources of bias.”

- If it is about health care utilization: There is an abundance of literature on health care utilization (numbers of consultations) in relation to socioeconomic differences, practice variation and qof. So if that is what this is about, the results are far from new.

Author reply: As above, our focus is on the frequency of codes occurring in the record, which may be different to utilisation. We discuss this issue further in the comment below.

- If it is about coding habits, the authors should give us more information about the link between health care and HER record keeping. Is it possible in the EMIS EHR system to record consultations without diagnosis codes? It possible that codes have been recorded without contacts/consultations? can there be contacts/consultations without codes. The methods section suggests that this is possible, suggesting the paper is about coding habits rather than consultations. But if it is, why not simply calculate diagnosis code / consultation ratio's.

Author reply: We have added more detail to the introduction regarding how codes can be entered. As above, it is possible for a consultation to be recorded without codes, or with any number of codes. Diagnostic codes may also be coded in the absence of a consultation, which may happen, for example, if retrospectively, a clinician wishes to add in a pre-existing diagnosis, or if communication is received from secondary care. We have also clarified the differences between payments between primary and secondary care, highlighting that there is no direct incentive for coding in primary care (except via QOF).

Because codes can occur outside of consultations, we cannot simply calculate the ratio of diagnosis codes to consultations. Furthermore, a ratio might be misleading, as we would still not know whether there were additional conditions that a patient presented with that were not coded (e.g. where patients are presenting with multiple complaints). If a diagnostic code is entered in the record, but not during a consultation, it is still relevant to the overall health status of a patient, and as such are important to include in any predictive model (as is typical in NLP modelling scenarios). We have added to the second paragraph of the introduction to explain the importance of considering both sets of codes in our context.

As discussed in answer to the first point, we also conducted a sensitivity analysis adjusting for total consultations partly to address the impact of total consultations on our findings, and have highlighted this further in discussion.

- One other problem with this paper is that there is no clear description of the problem addressed and why it is important.

Author reply: we have rewritten the introduction to more explicitly address the link to NLP studies and methods that use code sequences and the potential bias that this may introduce. We have also changed the title of the paper to better reflect the relevance to code sequences.

- A clear research question is lacking. As are clear statements regarding the hypotheses to be tested.

**Author reply: we have updated the introduction to more clearly state the problem that the paper seeks to address. The final paragraph has been rewritten as:
“Analytical methods using temporally-ordered code sequences in the EHR may therefore be susceptible to biases in the frequency of codes entered following diagnosis, potentially resulting in models representing some people better than others. Awareness of the factors influencing the frequency of codes may help researchers using NLP methods by informing adjustment or sensitivity analyses. This study aims firstly to compare the frequency of repeated codes after diagnosis for a common set of LTCs. Secondly, we aim to determine whether the frequency of codes varies according to i) disease inclusion in QOF, ii) GP practice, iii) patient socio-demographic characteristics, and iv) calendar year of diagnosis.”**

- The paper lacks a clear description of the types of study or study purposes for which the coding issue is relevant.

Author reply: in the rewritten introduction, we have more explicitly stated that this study has particular relevance for researchers using code sequences, such as with NLP methods. The amended title also reflects the relevance to code sequences.

Minor comments:

• Variation in referral rates is likely to impact the occurrence of codes/consultations.

Author reply: we agree this is likely to impact on first incidence of a code in the record where a patient requires a diagnosis in secondary care, but is likely to be less for codes recorded after first incidence as we do here.

• What are the implications of the study for Emis?

Author reply: we believe that the findings generalise more widely to those using EHR systems rather than EMIS specifically. We have added a paragraph to ‘Implications for clinical practice’ to address the implications for coding quality in EHRs and provided additional references.

• In several places it is suggested that patients are actually diagnosed in general practice. Very often this is not true. For many diseases patients are exclusively diagnosed in hospital.

Author reply: This may be the case for several LTCs, and we have added text to both introduction and methods (‘Disease definitions’) to highlight that diagnostic codes may be entered in the EHR following a diagnosis made in secondary care. We have also stated in discussion under ‘GP practice’ that practices may code “correspondence from specialist services” and that there may be differences between how practices code this information.

• Disease coding incentives: are they not incentives for providing care rather than coding.

Author reply: This was the objective of QOF, but there is evidence that QOF impacts on both. We discuss this in ‘QOF and non-QOF conditions’ in the Discussion and provide reference 14

(apologies as in the previously submitted version, and incorrect reference appeared here). We also highlight that it is difficult to assess whether the finding of higher code frequency for QOF conditions relates to more contacts or higher likelihood of coding and have made some amendments to the text in this section here to clarify.

- Selection of people who did not move house for two years, and who remained in the same practices. This will result in bias in the data studied. Has this effects on the results of the study?

Author reply: It is standard for many analyses of disease incidence using CPRD data to include people with at least 1 year of follow-up, to reduce bias in the overestimation of incidence from patients registering because of having a condition. We have explained this in the limitations, along with a reference (17) to a previous study showing incidence declines to baseline by 1 year. This may introduce a small amount of bias towards inclusion of older patients, but previous work from our team on CPRD data shows the difference in the cohorts to be small.

- What about completeness of ethnicity data. Is this recorded in EHRs? How complete and reliable is that information?

Author reply: we have added to 'Data source' in Methods the additional demographic variables we used, including ethnicity, and provided a reference (19) that indicates it aligns well with other NHS and national level data sources for the UK.

- Not made clear how qof is related to coding rather than to consultations.

Author reply: This should now be clearer with the amendments stated above to the section on 'QOF and non-QOF conditions'

- I don't understand how to interpret the scatterplots. If one disease has more coded diagnoses in a subsequent year after first incident consultation than another code. What does that mean?

Author reply: The scatterplots indicate that there is a strong correlation between code frequency for QOF and non-QOF conditions at a practice level. We have added to the text: 'indicating that those practices with high code frequency for QOF conditions also had high code frequency for non-QOF conditions.' We also discuss this further in 'QOF and non-QOF conditions' in the Discussion, and have added the reference here to Figure 1.

All in all, this may in the end turn out to be a very interesting paper with very relevant analyses, and I would advise the authors to solve the problems described and resubmit. Electronic health records data are currently being used for very many purposes. Epidemiological studies, health care utilization studies, health care quality studies, access to care studies etcetera. Possible biases might render the data unfit for some or all of these purposes and papers addressing such biases are important. All too often researchers tend to think that the data speak for themselves. I would like to encourage the authors of this paper to review their work carefully and then resubmit.

Author reply: Thank you for your helpful suggestions, which have given us the opportunity to revise the manuscript and for readers to better understand the relevance to their own research.

Reviewer: 2

Dr. Benjamin Ranard, Columbia University Irving Medical Center

Comments to the Author:

In this interesting study, the authors analyzed the diagnostic codes from a large group of outpatients in England. The stated rationale for the study was to identify factors affecting diagnostic coding of new diseases in order to inform future studies of sequences of diagnostic codes given to individual

patients. The authors showed that the frequency of diagnostic coding for new diseases was associated with patient sociodemographics, incentivization by the Quality and Outcomes Framework (a financial incentive program), GP practice, and the onset of COVID-19. This study has the potential to be useful in the future both for analyzing sequences of diagnostic coding and for health services research in general that aims to take advantage of diagnostic-code based data.

Author reply: Many thanks for your comments and helpful suggestions which we have addressed below.

Major Comments

1. Incident diseases are diagnosed by allowing at least one full year of practice registration before incident disease diagnosis. However, younger people (say 25-year-olds) seem less likely to visit their GP at least annually. Patients who visit their GP less frequently are more likely to be diagnosed with an incident disease when they do eventually visit their GP (since all of their prior comorbid diseases haven't been recorded in the study period). Am I correct in thinking that diseases associated with patients who visit their GP infrequently are more likely to be diagnosed as an "incident disease" when they do finally visit their GP?

Author reply: The GP record includes all retrospective clinical codes, and these should be transferred where a patient moves between practices. If, for example, a patient presents to a GP with a long-standing diagnosis which isn't already on the record, then this can be coded along with the date of observation (i.e. diagnosis) which is distinguished from the date of data entry. We have used the observation date throughout in preference to the entry date. Therefore, a pre-existing condition would not be counted as 'incident' in our study. Although this may not always be recorded accurately, we would not expect a large differential effect in certain groups. We have added to Methods, 'Disease definitions' to clarify that we used the date of observation, rather than date of entry. We used one year of practice registration as this is common in CPRD studies, as inclusion of those recently registered can lead to a spike in incidence rates. We have added a discussion and reference (17) in Methods and limitations.

2. Do we already know from other literature, or is there any analysis you can do, that would tell us how representative these diagnostic codes are of clinical reality? For example, is your incidental rate of diabetes using diagnostic codes at all similar to a more gold standard known rate of diabetes incidence in England? It would be helpful to know how well these incident rates of diagnostic codes are associated with the clinical reality.

Author reply: there are existing studies that have shown good agreement between CPRD and both clinical validity studies and national statistics, but these tend to depend on the presence of 'any' code, rather than giving an indication as to the validity of subsequent codes in the record. We have brought this explanation into the introduction, to highlight the comparability of the EHR prevalence data to other sources along with a reference: "Although previous studies of EHR data in England have shown the prevalence of many long-term conditions (LTCs) to be comparable to those from national statistics, these are often calculated based on the presence of a single diagnostic code.⁸"

Minor Comments

1. In the United States, diagnostic codes are mainly used as billing codes. Additional codes are added to encounters to justify the complexity of the medical visit for billing purposes. For providers in an Accountable Care Organization or some sort of value-based payment model, documenting additional codes may be incentivized to ensure providers receive "credit" for the patient's level of sickness or comorbidity. Additionally, ICD codes may be necessary for prescribing medications. Adding a line or two about how/why providers code in England (besides for QOF incentives) would be useful background for international readers.

Author reply: thank you for this suggestion and we agree this is important to highlight. We have introduced this in the third paragraph of the Introduction and the contrast to secondary care:

“Unlike in secondary care, where diagnostic coding directly impacts on payments, General Practice in England receives funding primarily through capitated payments based on the size of the registered population¹¹ with no direct financial incentive for code entry during a consultation. However, around 10% of funding comes from the Quality and Outcomes Framework (QOF), introduced in the National Health Service for GPs in 2004.”

2. Page 6 line 19: what is a research acceptable patient? Is this a patient who has indicated their medical record can be used in research?

Author reply: This is a term defined by CPRD based on certain quality criteria (e.g. registration date not being recorded before date of birth). We have modified the text to make this clear, and added a reference which explains the criteria.

3. Page 6 line 19: what is required to maintain “a continuous period of registration at a GP practice?” Does this mean the patients visited their GP at least in a certain frequency or is it more like the patient was signed up for a GP as of 1/2014 and they never removed themselves from the patient panel?

Author reply: In CPRD we have a date of registration and date of deregistration. Patients are deemed to be continuously registered if they have not deregistered, irrespective of whether they presented. We have added a clarification in Methods ‘Data Source’.

4. Appendix page 2. Can you add a justification for: “We reviewed the codes in these lists, and made amendments to the code lists for diabetes, to remove Type 1 and Type 2 codes from the other/unspecified code list.”

Author reply: We have modified the text in the appendix to add a clarification: “We reviewed the codes in these lists, and made amendments to the code lists for diabetes. The ‘other/unspecified’ diabetes code list contained codes specific to both Type 1 and Type 2 diabetes, and we removed these to ensure the list included only codes where a more specific Type 1 or Type 2 diagnosis was not stated.”

5. Page 6, line 55-56: Did the two primary care clinicians have perfect agreement or what was the Kappa? Not essential, but it would be nice to know the agreement or whether consensus was used.

Author reply: unfortunately, we did not capture this information when the assignment was done and the review was not done totally independently – an initial assignment was completed by TB, and then reviewed by DS. There were three codes where we then had further discussion and then agreed on the assignment.

6. Page 10: can you add in the appendix a “table 1” that compares the sociodemographic characteristics of patients included in the study with the 50% of patients who were excluded due to not having one incident disease in the study period? Is it possible to add a few clinical characteristics to both groups (x% had a diabetes code, y% had a heart failure code, etc)? This would not be incidence but whether the patient has ever had a code in order to get a sense of the patient population.

Author reply: Thanks for this suggestion. We have made some amendments to Table 1 to present the total number of pre-existing QOF and non-QOF conditions at the study start date. We have also provided in the appendix (Table A2) the frequency and percentage of all pre-existing diseases (with small counts for some diseases suppressed). In Table A3 we provide the age and sex break-down for the ineligible cohort, showing these patients tend to be younger and more likely to be male. We have added to the first paragraph of results to reflect these additions:

“Of patients with pre-existing conditions developed before the study start date, 31.6% had one or more QOF conditions, and 71.3% had one or more non-QOF conditions. Hypertension was the most prevalent pre-existing condition (24.1%), and the frequency of all pre-existing conditions are shown in the appendix Table A2. The 3,060,391 patients who were not eligible (as they did not develop an incident disease over the study period), were more likely to be younger and more likely to be male than those eligible (appendix Table A3).

7. Page 11: The Figure 1 legend is erroneously showing up on my PDF on page 11. And figure legend 2 and 3 are showing up on page 13 of PDF.

Author reply: we kept these in to indicate the position as it should appear in the text as per the BMJ Open submission guidelines and we can amend as the editors prefer.

8. Page 15 line 14 looks like it may be a typo.

Author reply: thanks for spotting this – corrected.

VERSION 2 – REVIEW

REVIEWER	Ranard, Benjamin Columbia University Irving Medical Center, Division of Pulmonary, Allergy, and Critical Care Medicine, Department of Medicine
REVIEW RETURNED	31-Jul-2023

GENERAL COMMENTS	Thank you for the revisions. I think the new version more clearly articulates the research question and results. A couple minor comments: Page 6 lines 59-60: "The date of the event ('obsdate') was used, in preference to the date the code was entered." I believe the estimated date of the diagnosis was used and not the date the code was entered into the EHR. The quoted wording in the manuscript is still a little confusing to me. Page 7 lines 9-15: "We excluded conditions based only on laboratory results or anthropometric measurement codes as these may have different characteristics of coding frequency. As a result, measures of raised cholesterol used in the original CALIBER study were excluded. We also excluded BMI and eGFR measurements but included the diagnostic codes for obesity and Chronic Kidney Disease." Why did you exclude cholesterol, BMI, eGFR but include CKD? CKD is mostly based off of historical creatinine and eGFR measurements. Type II diabetes is mostly based off of A1c for that matter. Not a major concern, but I didn't understand that quoted text. Figure 1: should same scale be used for each axis to give an accurate visual representation of slope?
--

VERSION 2 – AUTHOR RESPONSE

Reviewer comments:

Reviewer: 2

Thank you for the revisions. I think the new version more clearly articulates the research question and results. A couple minor comments:

Page 6 lines 59-60: "The date of the event ('obsdate') was used, in preference to the date the code was entered." I believe the estimated date of the diagnosis was used and not the date the code was entered into the EHR. The quoted wording in the manuscript is still a little confusing to me.

Author reply: thank you for highlighting this. We have amended this to "The date that the event occurred ('obsdate')..."

Page 7 lines 9-15: "We excluded conditions based only on laboratory results or anthropometric measurement codes as these may have different characteristics of coding frequency. As a result, measures of raised cholesterol used in the original CALIBER study were excluded. We also excluded BMI and eGFR measurements but included the diagnostic codes for obesity and Chronic Kidney Disease." Why did you exclude cholesterol, BMI, eGFR but include CKD? CKD is mostly based off of historical creatinine and eGFR measurements. Type II diabetes is mostly based off of A1c for that matter. Not a major concern, but I didn't understand that quoted text.

Author reply: We meant to acknowledge that these were specific decisions based on the CALIBER code lists. For some conditions in CALIBER, the diseases are coded only based on lab measurements (e.g. raised cholesterol), whereas for others there are a mix of measurements and diagnostic codes (CKD and obesity) and we used only those diagnostic codes. We have rewritten these lines as: "In CALIBER, conditions related to raised cholesterol or triglycerides are based only on laboratory results, rather than diagnostic disease codes. We excluded these conditions given that laboratory measurements may have different characteristics of coding frequency. Likewise, for obesity and Chronic Kidney Disease, we used the diagnostic codes included in the code lists, but did not include BMI and eGFR measurements."

Figure 1: should same scale be used for each axis to give an accurate visual representation of slope?

Author reply: we did originally try this, and agree in general it is preferable to use the same scale. However, this gives the appearance of the points being bunched tightly together, and does not give a good visual representation of the variation about the line. We have retained as separate scales, but highlighted this in the footnote.